# A Review of Ship-to-Ship Interactions in Calm Waters

**Claire DeMarco Muscat-Fenech** [1],*, **Tonio Sant** [1], **Vito Vasilis Zheku** [2], **Diego Villa** [2] **and Michele Martelli** [2]

1    Department of Mechanical Engineering, University of Malta, MSD2080 Msida, Malta
2    Department of Electrical, Electronics and Telecommunication Engineering and Naval Architecture, (DITEN), University of Genova, Via Opera Pia 11a, 16145 Genova, Italy
*    Correspondence: claire.demarco@um.edu.mt

**Abstract:** The hydrodynamic interaction between two or more ships in harbours or inland waterways is a classical maritime engineering research area. In ship manoeuvring practice, ship masters try to determine the speed and gap limit when a ship is passing or encountering others, particularly in confined water ways. This requires an accurate prediction of the interaction force acting on both ships. The pioneer experimental studies showed that the interaction could lead to a very large yaw moment and this moment is strongly time-dependent, which could make the ships veer from their original courses, leading to collisions. Based on the findings on experimental measurements, some empirical formulas are proposed in the literature to predict such interaction forces. However, these formulas could provide a satisfactory estimation only when the ship speed is quite high, and the water depth is shallow and constant. Numerical simulation overcomes this issue by simulating the ship-to-ship problem by considering the effect of the 3D ship hull, variable water depth and ship speed. Numerical simulation has now become the most widely adopted method to investigate the ship-to-ship problem. In the present study, the development of the methodologies of ship-to-ship problems will be reviewed, and the research gap and challenges will be summarized.

**Keywords:** ship-to-ship interaction; experimental model interactions; overtaking manoeuvres; head-to-head encounters; lightering operations; RANS solver

## 1. Introduction

A sea faring vessel moving at constant forward speed in water will encounter a three-dimensional pressure distribution around its hull surface as a result of the uneven velocity distribution induced by the relative motion between the vessel and water. A positive pressure zone is generated in the bow (due to the stagnation pressure) and stern regions (pressure recover effect), whereas a low-pressure region develops on both sides of the hull, as indicated in Figure 1. The low pressure (suction) regions result from the higher velocities, in accordance with the Bernoulli law. In open sea conditions, the complex pressure distributions depend mainly on the hull geometry, ship speed, sea depth and sea state. When the vessel is moving in a confined channel, for example in a canal or river, then the channel dimensions influence the pressure distribution across the hull. The ship-to-ship interactions are considered in calm waters under the influence of a wind speed of Beaufort number 0 and wave height 0 m, on a mirrorlike sea.

In the presence of heavy traffic, lightering operations and harbour manoeuvres, ship-to-ship encounters are common. The ship-to-ship interaction may induce significant hydrodynamic loads that may negatively affect the ability to manoeuvre the vessels in a timely manner and avoid the risk of collision. Considering the pressure distribution around each vessel (as reported in Figure 1), a repulsive force is experienced when two bows become close to one another. In the following figures, the main cases of ship-to-ship interaction found in the literature are shown. For a better understanding of the signs of the forces and moments induced by the ship-to-ship interaction, a scheme with the relevant coordinate

system is presented in Figures 2 and 3. The same notation used by Lataire et al. [1] is being presented, where subscripts STBL and SS stand for "ship to be lightered" and "service ship", respectively.

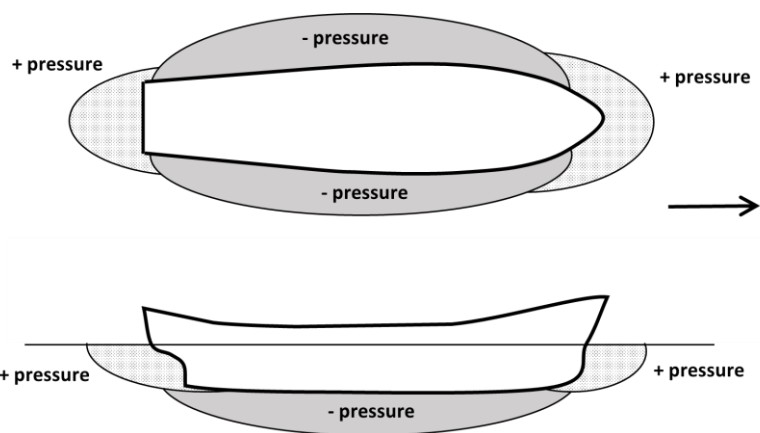

**Figure 1.** Pressure distribution around the hull of a ship in forward motion.

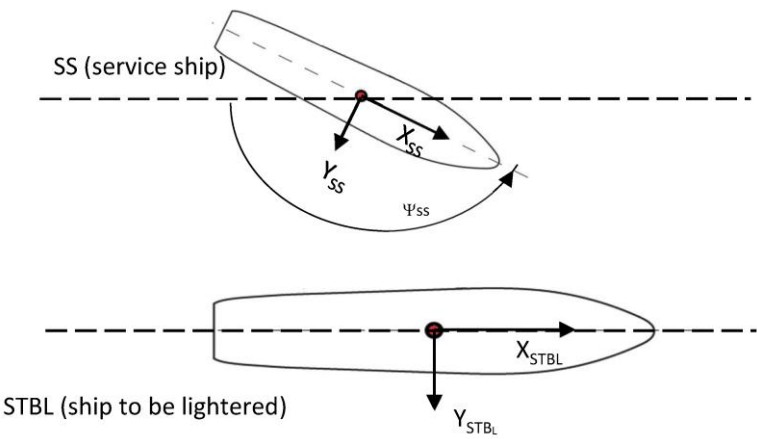

**Figure 2.** Coordinate system.

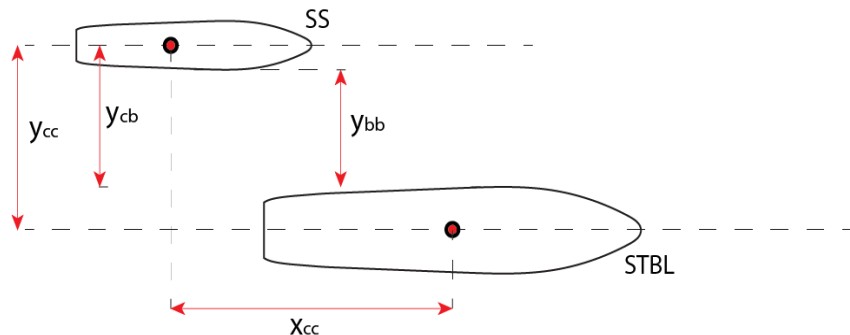

**Figure 3.** Pressure distribution around the hull of a ship in forward motion.

The ship-to-ship interaction notations, for the ships of beam $B$, found in the literature and illustrated in Figure 3 are:

$\xi$ [-]—longitudinal distance between the ship's midships sections ($x_{cc}$) divided by a reference ship length, being zero when both midship sections are aligned.

$y_{bb}$ [m]—lateral distance between ship sides

$y_{cb}$ [m]—lateral distance between own ship centre line and side of target ship:

$$y_{cb} = y_{bb} + \frac{B}{2}$$

$y_{cc}$ [m]—lateral distance between ship centre lines
$\eta$ [-]—dimensionless lateral distance, defined in literature in two different ways:

- Vantorre [2]:

$$\eta = \frac{y_{cc}}{\min(B_{STBL}, B_{SS})}$$

- Opheim [3], De Decker [4]:

$$\eta = \frac{y_{cc}}{\frac{1}{2}(B_{STBL} + B_{SS})}$$

*Head-to-head encounters.* Figure 4 presents a basic explanation of how ship-to-ship interaction loads are generated when two vessels are in a head-to-head encounter along a channel. The pressure distribution also induces a yawing moment, causing the vessels to rotate in opposite direction, pushing each bow towards the channel bank (Stage 1, Figure 4). As soon as the bow of one vessel approaches the stern of the other, an attractive force is then encountered, leading to a change in the direction of the yawing moment and favouring realignment with the channel centre line (Stage 2, Figure 4). Once the stern of each vessel comes into close proximity of the mid region of the other vessel, an attractive force is experienced (Stage 3, Figure 4), resulting in a reversal in the direction of the yaw moment, which causes the bows to head towards the channel banks. A final change in the direction of the yawing moment is encountered as the two sterns pass by one another (Stage 4, Figure 4).

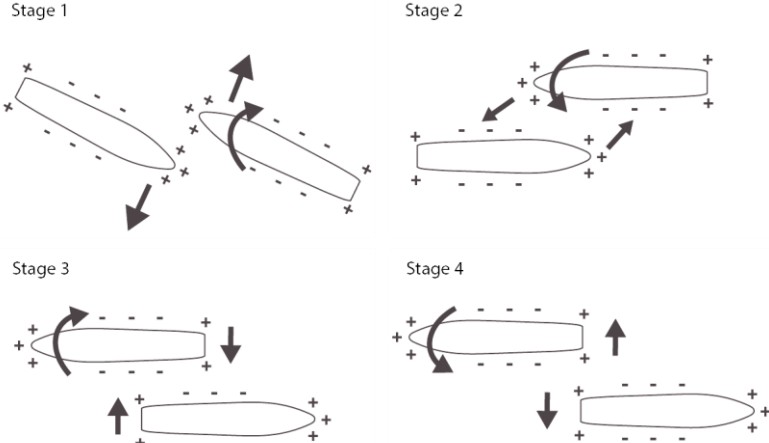

**Figure 4.** Interaction loads between two vessels during a head-on encounter.

Figure 5 shows the typical variation in the yawing moment induced on a ship's hull during a head-on encounter corresponding to the motions described in Figure 4. Positive values represent the situation when the two bows are pushed away from the channel centre line, and negative values when the sterns are pushed. The previously mentioned stages are indicated in the figure.

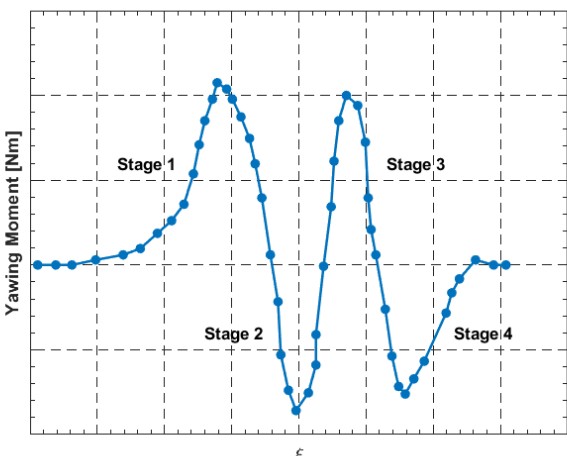

**Figure 5.** Typical variation in the induced yawing moment on a vessel during a head-on encounter.

*Overtaking encounters:* When two ships overtake each other, different forces and motion can be expected as the positive and negative regions around the vessels interact. Figure 6 explains the interaction loads during overtaking. In stage 1, the repulsive forces generated by the pressure distribution around each vessel induce a yawing moment, when the two positive pressure regions of the stern of the leading vessel and fore of the trailing or passing vessel causing the vessels to rotate towards the channel bank. In stage 2, the overtaking vessel advances and its bow is attracted towards the other vessel's negative pressure region midship, and the force on the overtaken vessel stern induces a yawing moment, pushing the vessel's bow towards the bank. In stage 3, as soon as the passing vessel passes the midship of the overtaken vessel, the yawing moments are reversed. During the last stage, 4, the stern of the passing vessel pushes the bow of the overtaken vessel towards the bank. A typical variation in the yawing moment induced on a vessel during an overtaking manoeuvre is shown in Figure 7.

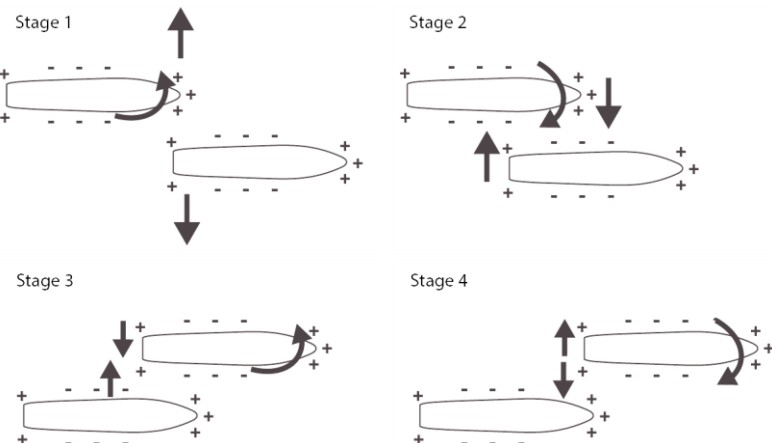

**Figure 6.** Interaction loads between two vessels during overtaking.

The most influential parameters affecting the magnitude of interaction loads in calm waters include the (1) lateral distance between the vessels, (2) position of the vessels with respect to each other along the direction of motion, (3) hull geometries, (4) hull draught, (5) speed and acceleration, (6) water depth and (7) secondary influences from the propellers and rudder.

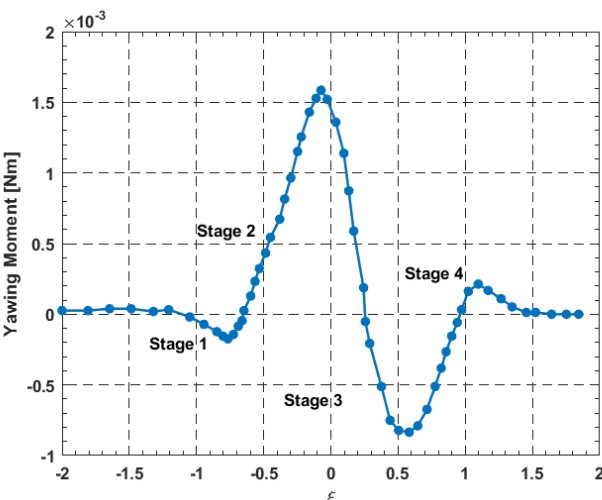

**Figure 7.** Typical variation in the induced yawing moment during overtaking.

Various experiments on ship models under controlled conditions in wave basins have been carried out to study the underlying phenomena of ship-to-ship interaction. The studies mainly focused on the measurement of the three contributions on the horizontal plane: the surge force, the sway force and the yawing moment. A major challenge with conducting ship-to-ship interaction studies is the large number of variables that affect the induced loads, thus leading to a large combination of variables with many experiments. Furthermore, to investigate the influence of relative speed between two vessels, two independent carriages would have to be installed in the test facility. Facilities with this ability are fairly limited. The following section reviews a selection of experimental studies documented in the open literature.

The objectives of the work presented is to firstly present the pioneering experimental studies on strongly time-dependent ship-to-ship interactions, and the subsequent possible large yaw moments which deviate the ships from their original course resulting in possible collisions; to describe the available literature's empirical formulae to predict the resulting interaction forces, which provided a reasonable estimation when the ship speed is high, and the water depth is shallow; to review the methodology and numerical simulations by considering the effect of 3D ship hull, water depth and ship speed, which improves the ship-to-ship interaction predictions; and, finally, to present the research gap and challenges.

Another important aspect of the correct evaluation of the interaction forces is their importance during the design procedure of the mooring systems. Recent studies in this field aim to improve the conventional mooring systems, in order to help the industry, save material and, thus, overall project costs. Ja'e et al. (2022) [5] presents an optimisation procedure of mooring line design parameters for a turret-moored FPSO. The tool used was an in-house one named Mooring Optimization Tool for FSPO (MooOpT4FSPO). The results, consisting of static offset, free decay and hydrodynamic response were compared to published results (Montasir et al., 2019 [6]) showing good agreement.

## 2. Experiments on Ship-to-Ship Interaction

The first experiments studying ship-to-ship interaction have been reported by Gibson [7]. These involved measurements on screw-propelled models of different sizes, tested at distances pf up to 200 yards apart. One of the first published measurements has been those of Newton [8] and Muller [9]. Newton [8] performed tests for overtaking manoeuvres for deep water conditions and concluded that although the moment of the force created by the rudder can effectively counteract the interaction moment, under some occasions, the rudder force is insufficient to balance out the induced force resulting from ship-to-ship interactions, unless the ship is slightly yawed to produce a lateral force opposing the inter-

action force. Muller [9] conducted measurements for two ships meeting and overtaking in a narrow channel.

Remery [10] performed a series of tests in a ship model basin in the Netherlands to measure the mooring loads on a moored 1/60 scale fully loaded 100 MDWT tanker during the passage of a tanker. The size and speed of the passing tanker were varied, together with the distance between the vessels. The water depth was equal to 1.15 times the draught of the moored ship. For some tests, the stiffness of the mooring system was varied using a flexible system of linear elasticity. It was found that the loads induced by a passing ship on a moored vessel are proportional to the square of the speed of the passing ship. Furthermore, the stiffness of the mooring system was noted to have a considerable effect on the mooring forces. Smaller mooring forces were measured when the mooring system was stiff.

Vantorre et al. [2] carried out more extensive model tests for ship interaction. Load measurements were performed to evaluate the ship-to-ship interaction between two models in both head-to-head encountering and overtaking. The test facility included two independent carriages, allowing for high resolution measurements of the surge and sway forces and the yawing moment induced by the interaction of two vessels moving at different speeds. Four different hull geometries on a 1/75 scale were tested for speeds of up to 16 knots. It was observed that higher harmonics are introduced in the time history of the load measurements, as the length of the two models differs considerably. The forces and moments were also found to increase significantly as the draught of the target ship or the water depth were increased.

Opheim [3] and De Decker [4] conducted the same experiments by using the same ship models in a different experimental facility. The experiments focused on lightering operations, with the smaller model being a tanker at ballast draught and the larger one being a large tanker at a fully loaded draught. The measurements confirmed higher load coefficients at larger speeds and smaller lateral distances. Both experiments achieved very similar results for the sway force, yet significant disparities were observed in the case of the surge force. This was likely due to vibration issues that affected data quality.

Lataire et al. [1,11,12] used the same test facilities used by Vantorre [2] to conduct a captive model test program on a 1/75 scale for a ship lightering operation (Figure 8). Load measurements were carried out on the induced forces and moments on the service ship in close proximity of the ship being lightered. Measurements have been taken with both ships moving in the same direction with the same speed in deep waters. Automated systems enabled an average of 35 measurement runs to be conducted every day through the test campaign. Apart from steady state tests, during which the main parameters such as the speed and the longitudinal and lateral position were kept constant, dynamic tests were conducted involving a varying rudder, lateral distance and heading of the service ship. Three dynamic tests were undertaken: (1) harmonic sway tests, (2) harmonic yaw tests and (3) harmonic rudder angle tests. Further measurements conducted included those reported by Zhou and Larsson et al. [13] using the same models.

Sutolo et al. [14] conducted towing tank experiments to validate a potential flow code. The study focused on two vessels, a tanker and tug, travelling in parallel at a steady speed. Data was obtained for both shallow and deep-water conditions, with the models having a scale ratio equal to 1/25. No propellers were included. The tug was connected to the carriage through two strain gauges, positioned at the fore and aft to be able to derive the surge, sway and yawing moment acting on the vessel. The tug was allowed to heave, pitch and roll. On the other hand, the tanker model was rigidly fixed to the carriage and no measurements were taken to measure the hydrodynamic loads acting on it.

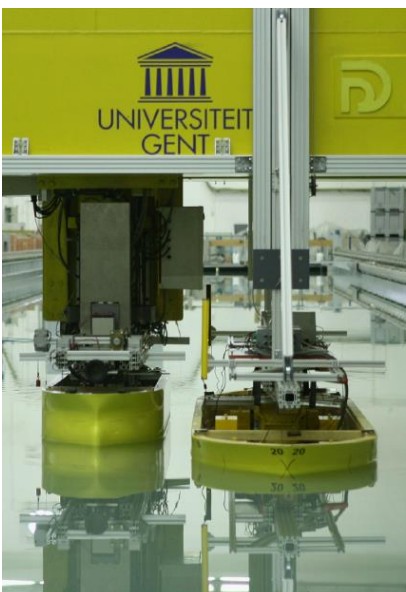

**Figure 8.** Experimental setup. Lataire et al. [1]. (Reproduced with permission from E. Lataire, [1]).

Denehy et al. [15] present results from physical scale model experiments to investigate the interaction forces and moments imparted on a berthed ship due to a passing ship. The study focused on examining the influence of blockage around the bow and stern around the berthed ship. The five different cases investigated with different bows and stern blockages are shown in Figure 9. The study was limited to shallow waters, with the depth-to-draught ratio not exceeding 1.054. The added stern blockage (Case D) was found to slightly decrease the peak negative surge force and the initial peak of the positive sway force. A reduction in the peak positive yaw moment and an increase in the peak negative moment were also observed. On the other hand, the presence of both bows and stern blockage (Cases B and E) led to larger peak to peak forces and reduced sway force and yaw moment.

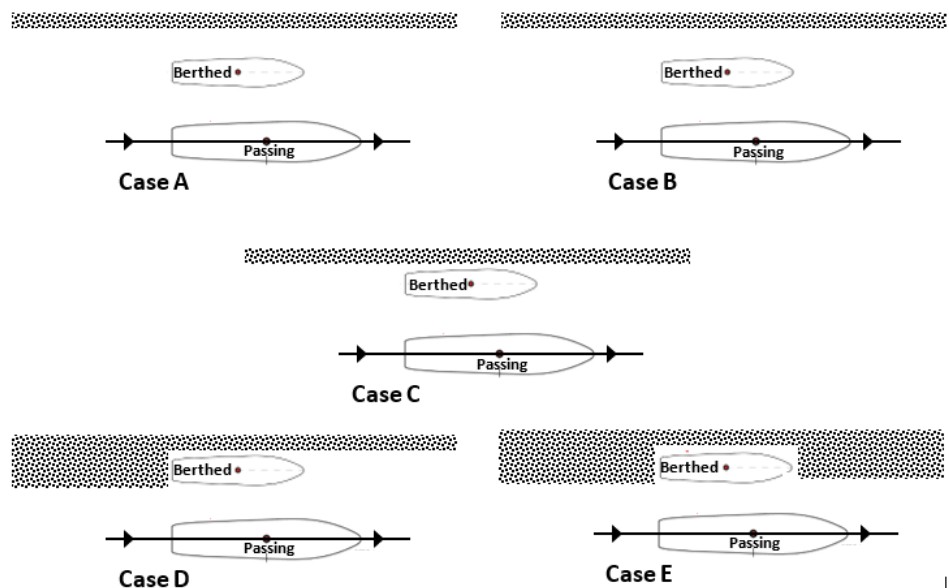

**Figure 9.** Case studies to investigate the role of bow and stern blockage. Adapted from Denehy et al. [13].

Yu et al. [16] presented the experimental results from towing tank tests on overtaking manoeuvres as part of a validated study involving CFD. The two models were identical and consisted of Series 60 hulls with a scale ratio of 1/80. One hull was kept fixed using

an aluminium cantilever arm, whereas the other hull was assembled to the towing tank carriage. The lateral forces on both hulls were measured, together with the drag of the overtaking hull.

Table 1 summarises the main experiments reported above, including the ship type being tested, facility used and the main scope. A common shortcoming is the lack of reporting of measurement uncertainty.

**Table 1.** Summary of ship-to-ship interaction experiments.

| References | Facility | Ships | Focus |
|---|---|---|---|
| Remery [10] | The Netherlands Ship Model Basin | • Tanker 1<br>• Tanker 2 | Mooring loads induced by a passing ship |
| Vantorre et al. [2] | Flanders Hydraulics Institute (Belgium) | • Bulk carrier<br>• Container ship<br>• Tanker<br>• Small tanker | Encounter and Overtaking manoeuvre |
| Opheim [3], De Decker [4] | Marintek (Norway) | • Tanker at ballast draught<br>• Large tanker at fully loaded draught | Lightering operation |
| Lataire et al. [1,11,12] | Flanders Hydraulics Institute (Belgium) | • Aframax<br>• VLCC | Lightering operation |
| Sutolo et al. [14] | FORCE Technology (Denmark) | • Tanker<br>• Tug | Parallel route at steady speed |
| Denehy et al. [15] | Australian Maritime College's Model Test Basin | • MarAd F bulk carrier | Forces and moments on a berthed ship due to a passing ship |
| Yu et al. [16] | Physical-model Testing Facility of Richmond Field Station (Berkeley, USA) | • Series 60 hulls | Overtaking manoeuvre |

The lack of reporting the uncertainties raises difficulties in validating computational models. It is well known that the accuracy of measurements is affected by the motion encountered by the vessels undergoing tests, both resulting from the degree of freedom allowed by the fixtures attaching the models to the tow tank carriage as well as from any vibration induced by the carriage itself during motion.

## 3. Empirical Models for Ship-to-Ship Interaction

Employing Vantorre's experimental results, De Decker [4] attempted to determine the applicability of Vantorre's results for lightering operations. The regression analyses were based on three dimensionless coefficients, $C_X$, $C_Y$, $C_N$, of surge, sway and yaw, respectively. The coefficients mainly depended on:

- The main dimensions of the models;
- The combination of the different ships;
- The longitudinal and lateral position;
- Ships' speeds.

The forces in the surge *(X)*, sway, *(Y)* and yaw moment, *(N)* were dimensionless:

$$C_X = \frac{X}{\frac{1}{2}\rho BTU_1 U_2} \tag{1}$$

$$C_Y = \frac{Y}{\frac{1}{2}\rho LTU_1 U_2} \tag{2}$$

$$C_N = \frac{N}{\frac{1}{2}\rho BLTU_1 U_2} \tag{3}$$

where $B$ is the beam (m), $T$ is the draught (m), $U$ is the speed (m/s), $L$ is the length (m) of the models, $\rho$ is the water density and $U_1$, $U_2$, are the velocities of the two ships.

The model experiments conducted by Vantorre [2,17] were undertaken in shallow water conditions, whereas in-service lightering manoeuvres normally occur in deep water conditions. Consequentially, De Decker [4] performed the lightering tests in Marintek, Norway, in deep water conditions, and states that as far as the tendencies are concerned, an overtaking operation in shallow water could be used to describe a lightering operation.

To compare the results, Decker [4] employed a 23rd degree polynomial curve fit to obtain the curves shown in Figure 10. Comparing the results clearly show, that for all cases, the amplitude of the forces and moments differ, and in the case of $C_X$, the trends are different.

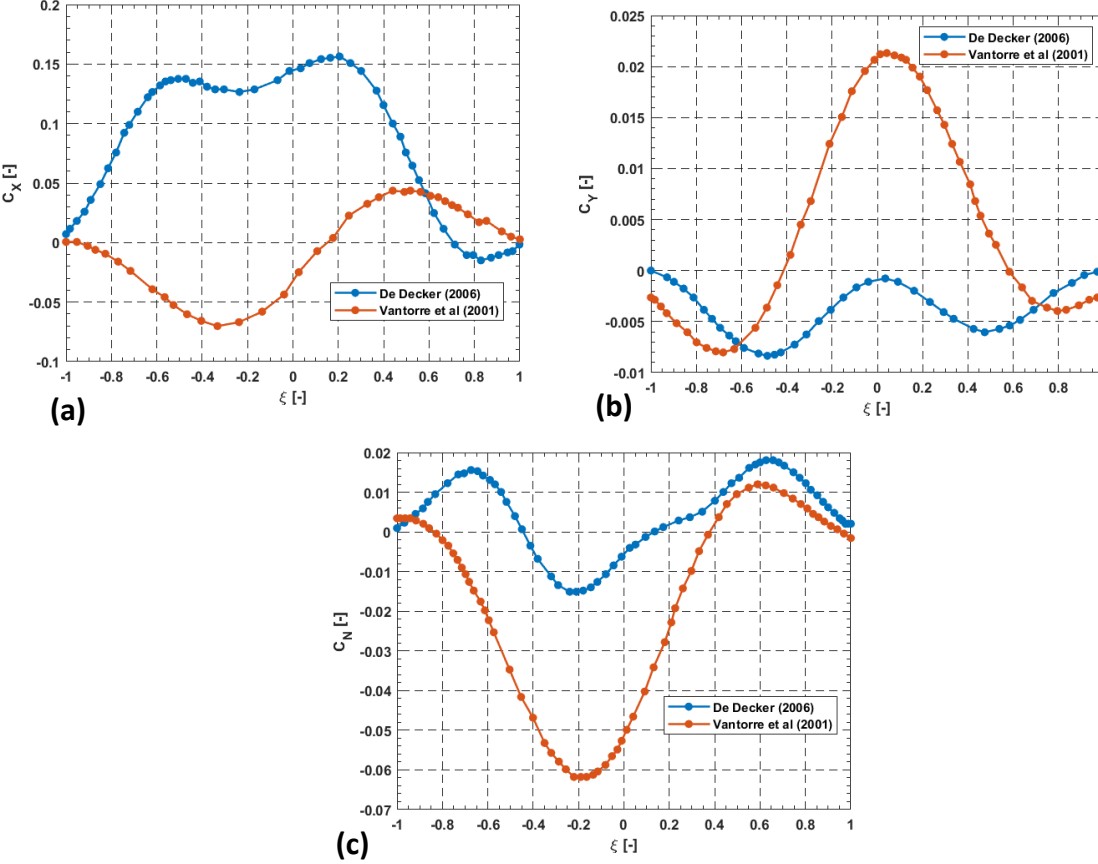

**Figure 10.** Dimensionless force and moments comparison; (**a**) $C_X$ for $\eta = 1.8$; (**b**) $C_Y$ for $\eta = 1.8$; (**c**) $C_N$ for $\eta = 1.8$. Adapted from De Decker [4], Vantorre [2,17].

An important point worth pointing in Vantorre's study [2,17] was that the hull had appendages, whereas both De Decker [4,15] and Opheim [3] used a naked hull form. Therefore, Vantorre's [2,17] results represent a better agreement with the real scenarios.

During the lightering operation, both draught and displacement of the two vessels change. These changes and the consequences were neglected in these studies. These effects of the draught and displacement change together with different kind of vessels that might be used for the lightering operations and constitute a possible investigation that might be interesting for future studies (De Decker [4]).

Lightering operations were based on experimental results conducted by Lataire et al. [11]. The proposed mathematical model takes into consideration a considerable number of relative longitudinal and lateral positions, loading conditions and forward speeds, studied in deep water conditions resembling true lightering operations. Initially, the expected extreme values were modelled, followed by a predefined shape function for the in-between values. The global model was based on the general trend of sinusoidal

functions for the surge ($X$) and the yaw moment ($N$), and the cosine function for the sway force ($Y$). The proposed mathematical models are given by Equations (4)–(6).

$$X_{STBL} = C_{Xmax} \frac{1}{2}\rho V^2 \frac{\nabla_{SS}}{Lpp_{STBL}} \left(\frac{T_{STBL}}{B_{STBL}}\right)^{C_T} \left(\frac{B_{STBL}}{y_{cb}}\right)^{C_y} \sin(\pi\xi_{SS})e^{-\xi_{SS}2} \tag{4}$$

$$Y_{STBL} = C_{Ymax} \frac{1}{2}\rho V^2 \frac{\nabla_{SS}}{Lpp_{STBL}} \left(\frac{T_{STBL}}{B_{STBL}}\right)^{C_T} \left(\frac{B_{STBL}}{y_{cb}}\right)^{C_y} \cos(c_\xi \pi\xi_{SS})e^{-\xi_{SS}2} \tag{5}$$

$$N_{STBL} = \frac{1}{2}\rho V^2 \frac{\nabla_{SS}}{y_{cb}} \left(\frac{T_{STBL}}{B_{STBL}}\right)^{C_T} \left(C_{Nsymm} \sin(\pi\xi_{SS})e^{-\xi_{SS}^2} + C_{Nasymm} e^{-(1+5.2\xi_{SS})^2}\right) \tag{6}$$

where $\rho$ (kg/m$^3$) is water density and $V$ (m/s) is ship speed. The exact values of the parameters $C_{Xmax}$, $C_{Ymax}$, $C_{Nmax}$, $C_Y$, $C_T$ and $C_\xi$ are not disclosed by Lataire et al. [1], even if a range of possible values is provided. The subscripts *STBL* and *SS* are explained in Figure 2; $\xi$ is the ratio of the longitudinal distance between the ships' midship sections to a reference ship length.

A good agreement was shown between the model test results and the mathematical model shown in Figures 11–13, showing the trends of the surge, sway and yaw, respectively.

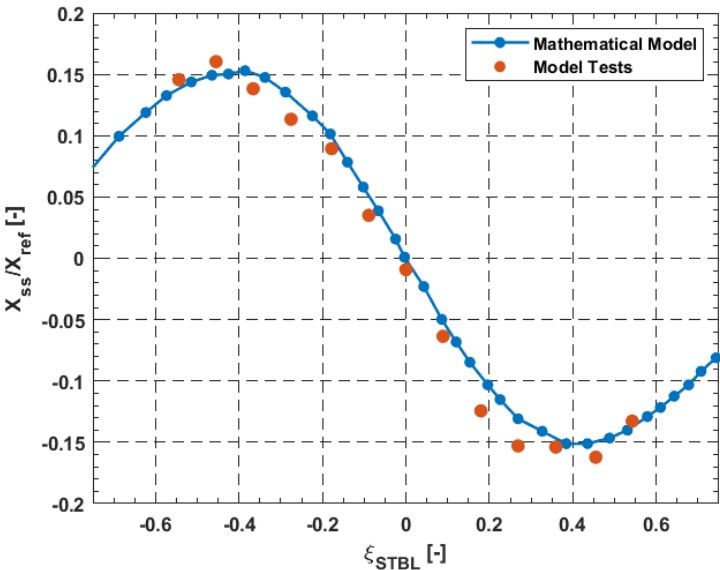

**Figure 11.** Comparison between the measured surge force and the result of the mathematical model; $y_{bb}$ = 10.0 m at 4.0 knot. Adapted from Lataire et al. [1].

As is usual with many mathematical models, some simplifications are adopted. The hypothesis of the symmetry of the peaks is not always verified by the experimental data, as well as the assumption that the maxima of each force/moment occur at the same value of $\xi$. As the mathematical models use sine and cosine, a bias is needed to shift the trend of the experimental data, in order to be symmetrical to the origin or to y-axis. The main benefit of these models, compared with others, is that of having a limited number of coefficients permits simplified and time-efficient simulation of ship manoeuvring. It should also be highlighted that some simplifications were made during the modelling.

Varyani et al. [18] proposed three new empirical models based on sway and yaw moment coefficients, previously evaluated by vortex distribution numerical techniques and slender body theory. The first two models are used for predicting the peaks of sway forces and yaw moments for both scenarios of head-to-head encounters and overtaking manoeuvres. The third model concerns the generic prediction of the sway force and yaw moment coefficients for two ships on parallel paths in encounter manoeuvres. Figure 14 shows the comparison between the results obtained from the two models.

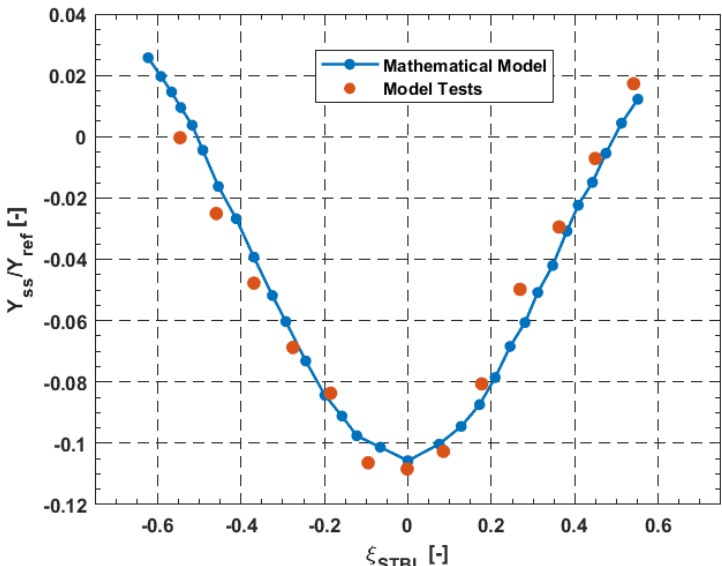

**Figure 12.** Comparison between the measured and modelled sway force; $y_{bb}$ = 10.0 m at 4.0 knot. Adapted from Lataire et al. [1].

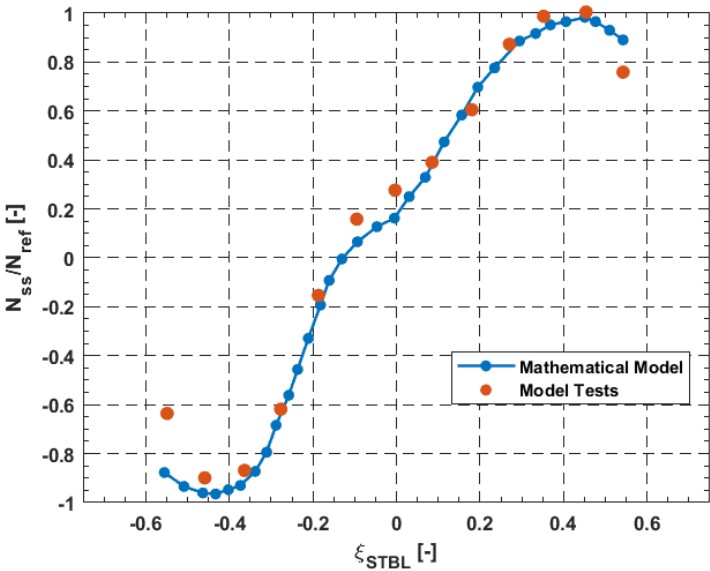

**Figure 13.** Comparison between the measured and modelled yaw moment; $y_{bb}$ = 10.0 m at 4.0 knot. Adapted from Lataire et al. [1].

Where the CF implies the non-dimensional sway force, the coefficient is calculated as follows for the Ship 1:

$$CF_1 = \frac{F_1}{0.5\rho U_1 U_2 B_1 D_1} \tag{7}$$

The ST′ non-dimensional stagger is calculated as:

$$ST' = 2.0 * \frac{ST_{12}}{(L_1 + L_2)} \tag{8}$$

For $U_1$ & $U_2 > 0$, $ST_{12}$ is equal to:

$$ST_{12} = (U_1 - U_2) * t \tag{9}$$

The non-dimensionalisation is such that the values $-1$, 0 and 1 refer to the bow–stern, midship–midship and stern–bow configurations, respectively.

To the best of the authors knowledge, no other methods or formulation are present in open literature.

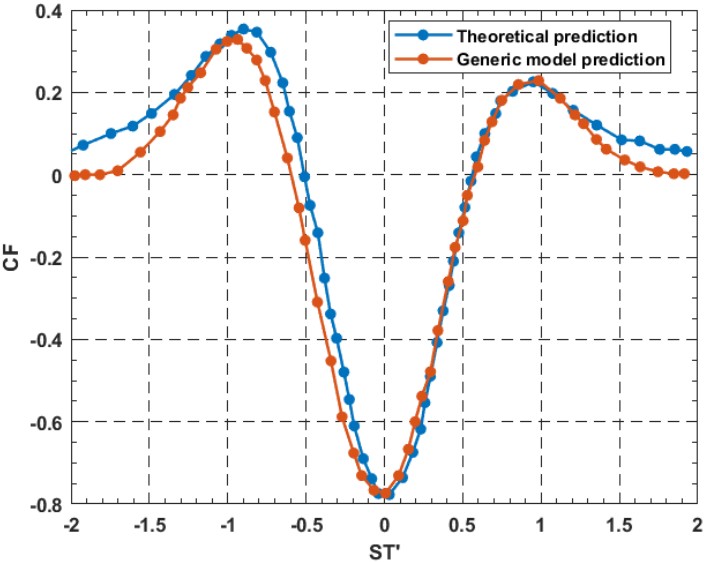

**Figure 14.** Comparison between the theoretical prediction and the proposed generic model by Varyani et al. [18] for the sway force coefficient during an encountering manoeuvre.

## 4. Numerical Methods for Ship-to-Ship Interaction

In the literature, the study of the ship-to-ship interaction via numerical methods can be divided in two main categories: viscous and potential flow methods. Initial studies on the ship-to-ship interaction were focused on seakeeping studies. Ohkusu [19] first introduced the analysis of the hydrodynamic forces on multiple cylinders in beam waves, calculating the response by the multipoles method and theory. Kodan [20], using the strip method, investigated the motions of adjacent floating structures in oblique waves.

Chen and Fang [21] investigated the hydrodynamic coupled motions problems between two vessels in waves. A 3D potential flow theory was used, where the potential is represented by a distribution of sources over the ship hull. The numerical calculations were made for two pairs of models, a barge-ship model and a Mariner-Series 60 model. The selection of these two specific pairs was made for comparison purposes. Data obtained from 2D methods can be found in the previous studies for the both barge-ship model (Kodan [20]) and the mariner-Series 60 model (Fang [22]). Moreover, Kodan [20] also provides experimental data, as well, related to the zero-speed condition. Numerical results obtained from the 3D method used by Chen and Fang [21] indicate that the forces calculated with the 2D methods are generally overestimated, especially around the resonance region, whereas a 3D approach shows good agreement with the experimental data. On the other hand, for the speed effect, the other pair was used as results obtained by 2D method exist already (Fang [22]). A reduction in the heave and pitch motion, caused by the sheltering effect, is not always experienced during a ship-to-ship interaction. From the study, it resulted that there is not always a reduction in the heave and pitch motions with the sheltering effect. It is clear that the reduced values obtained by the 3D method are more realistic from a physical point of view, as the standing wave between two ships is not completely trapped. Overall, 2D methods fail to estimate forces and moments induced by the ship-to-ship interaction when high forward velocities are involved, as the 3D effects become more dominant.

The significant advancement in computer power over the past decade allowed for Computational Fluid Dynamics (CFD) approaches to make considerable progress. Further-

more, the study of the ship-to-ship interaction started focusing on the manoeuvrability aspect. Increased interest was recently noted on operations that take place in the open sea, such as lightering and replenishment.

Zou and Larsson [13] studied ship-to-ship interaction during lightering operation using a steady-state Reynolds Averaged Navier-Stokes, RANS, solver. The software used for the numerical computations is SHIPFLOW, which contains a solver XCHAP based on the finite volume method. For grid generation, the overlapping grid technique was used. The flow field around each hull is covered by a body-fitted cylindrical H–O grid, whereas curvilinear O–O component grids were used for the purpose of describing the rudder geometries behind the two hulls. Cylindrical component grids represent the propeller discs behind the hulls, in order to apply the forces from the lifting line potential flow used to approximate the rotating propeller. A more detailed mesh arrangement can be found in Zou and Larsson [13].

The first part of Zou and Larsson's study aimed at extending the previous investigations of Sadat-Hosseini et al. [23,24] of the lightering operation in shallow water, utilising an unsteady RANS method. Zou and Larsson [13] applied the method to benchmark tests conducted by FHR. A comparison between the two numerical computations and the measured data shows that there is a good agreement between the two sets of computations. Differences were remarkable between the two CFD methods and the measured forces and moments and warrant further investigation is needed. In the second part, an attempt for systematic computations was made, focusing mainly on the influence of the lateral and longitudinal position between the two vessels. The lightering manoeuvre was approximated as a quasi-steady overtaking process, splitting it into several steps while modifying the relative longitudinal and lateral positions, as shown in Figure 15. In order to explain the forces and the moments acting on the hulls, the predicted pressure distribution was used. In the Figures 15 and 16, the axial velocity contours on the horizontal plane $z = 0$ at each $\Delta x$ step and the pressure distributions on the portside of the Aframax and on the starboard side of the KVLCC2 are presented, respectively.

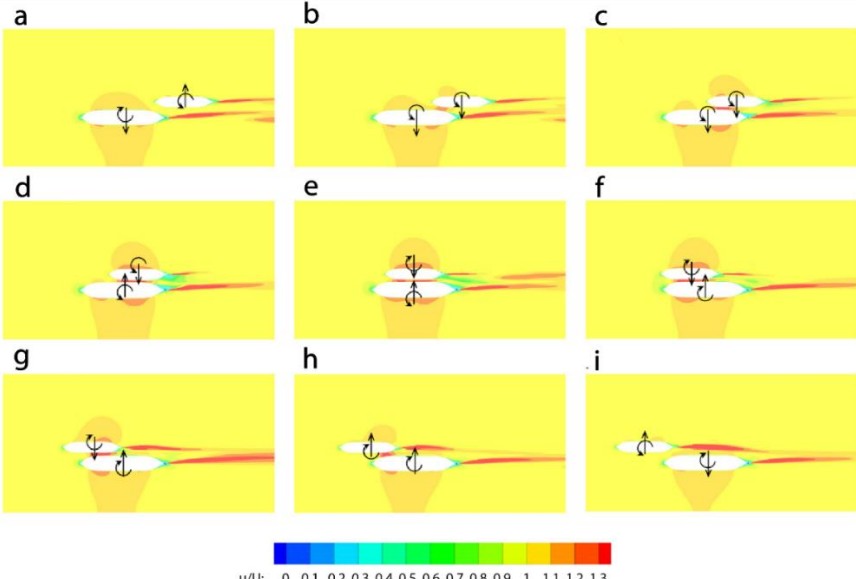

**Figure 15.** Example of axial velocity contours at the horizontal plane $z = 0$ against $\Delta x$, for $\Delta x/L_{Aframax} = -1.0, -0.75, -0.5, -0.25, 0, 0.25, 0.75$ and $1.0$, case (**a–i**) respectively. (Reproduced with permission from L. Larsson [13]).

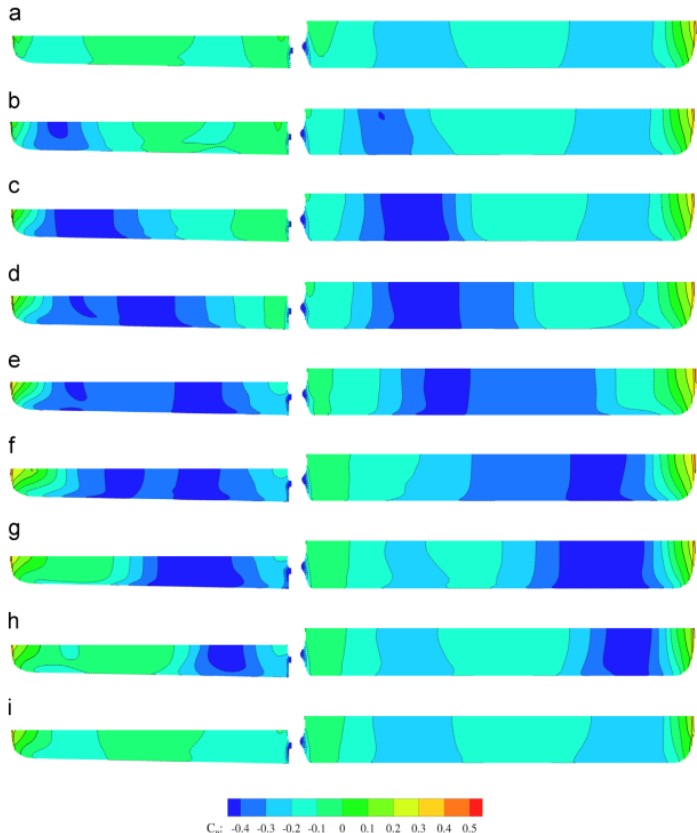

**Figure 16.** Example of pressure distributions on port side of the Aframax (**left**) and starboard side of the KVLCC2 (**right**) against $\Delta x$ for $\Delta x/L_{Aframax} = -1.0, -0.75, -0.5, -0.25, 0, 0.25, 0.75$ and $1.0$, case (**a–i**) respectively. (Reproduced with permission from L. Larsson [13]).

The cases presented in the previous Figures 15 and 16 are related to different $\Delta x/L_{Aframax}$, ratios of $-1.0, -0.75, -0.5, -0.25, 0, 0.25, 0.75$ and $1.0$. As previously mentioned, an other aspect under investigation was the influence of the lateral distance, $\Delta y$. Figures 17 and 18 show the axial velocity contours on the horizontal plane $z = 0$ at each $\Delta y$ step and pressure distributions, for $\Delta y/L_{Aframax}$, ratios of $0.648, 0.432, 0.324, 0.259$ and $0.233$.

The main conclusions of Zou and Larsson [13] were in good agreement with the results obtained during their former experiments. By altering the longitudinal distance between the two vessels, the resistance of the Aframax could increase or decrease, depending on its position with respect to the KVLCC2. On the other hand, the KVLCC2 always experiences a reverse change of resistance (Zou and Larsson [13]). At greater longitudinal distances between the vessels, a repulsive force will be generated between the two hulls, and when the vessels approach each other, attractive forces are generated. As far as the lateral distance between the two vessels is concerned, it was discovered that there will be an increase in the forces and moments when such a distance between the two hulls was reduced.

Yuan et al. [25] developed a 3D panel code, named Mhydro, based on Rankine source distribution to evaluate the hydrodynamic interaction between two ships on parallel courses. Both stationary and forward speed cases were considered as nowadays, lightering operations without forward speed are garnering attention. The results were compared to the published data obtained by commercial software and experimental results. For the stationary case, two scenarios were considered: Two Wigley hulls at the head sea condition; a Wigley hull and a rectangular box at the beam sea condition. On the other hand, for the ship-to-ship with forward speed case, a tanker-LNG ship model was studied and validated with experimental data from Ronaess (2002) [26].

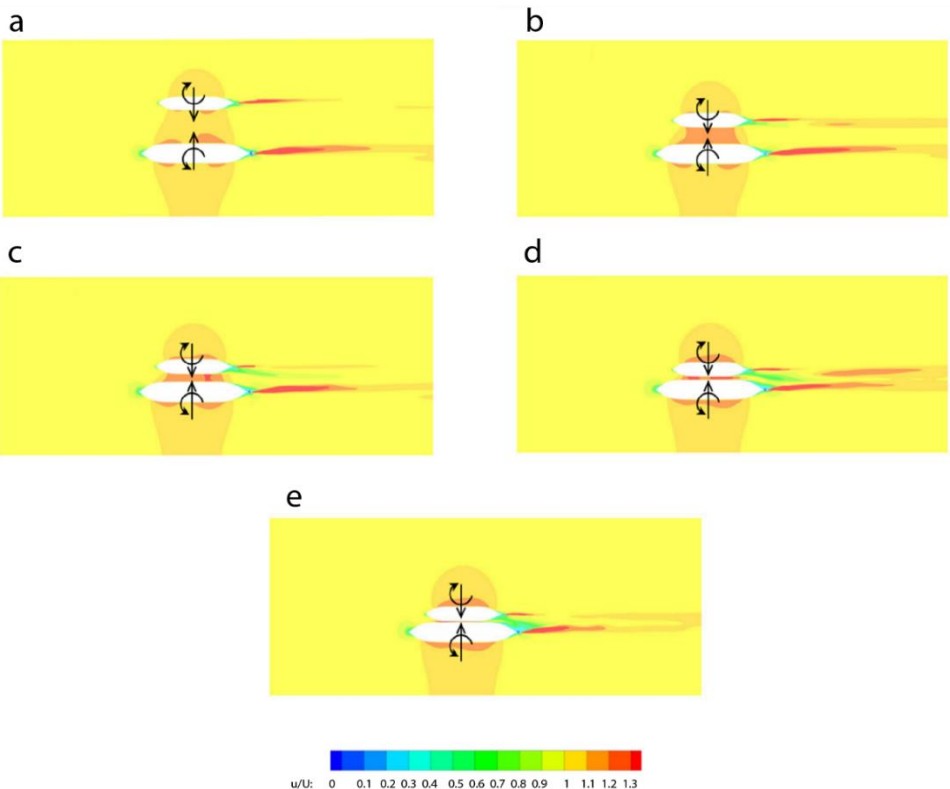

**Figure 17.** Example of axial velocity contours at the horizontal plane $z = 0$ against $\Delta y$, for $\Delta y/L_{Aframax}$, ratios of 0.648, 0.432, 0.324, 0.259 and 0.233, case (**a**–**e**) respectively. (Reproduced with permission from L. Larsson [13]).

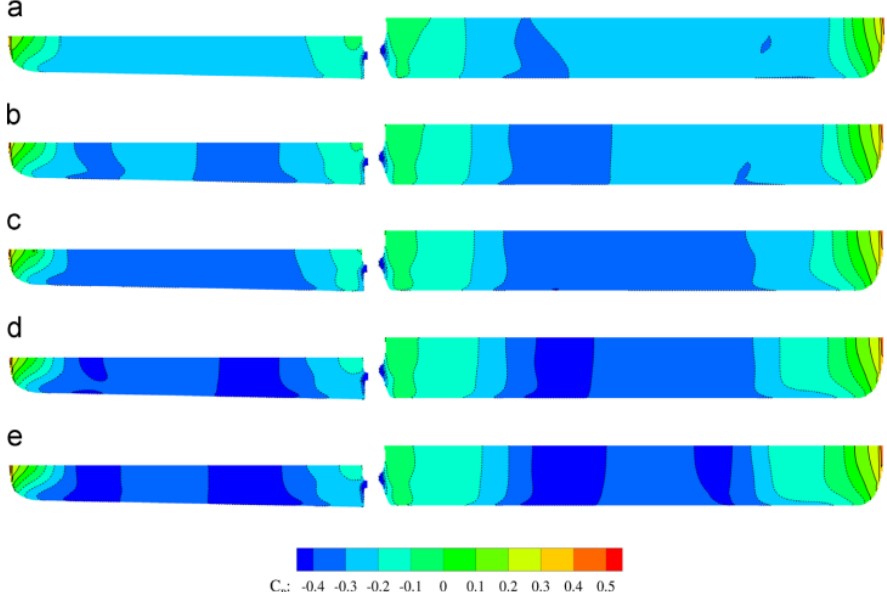

**Figure 18.** Example of pressure distribution on port side of the Aframax (**left**) and starboard side of the KVLCC2 (**right**) against $\Delta y$, for $\Delta y/L_{Aframax}$, ratios of 0.648, 0.432, 0.324, 0.259 and 0.233, case (**a**–**e**) respectively. (Reproduced with permission from L. Larsson [13]).

Figure 19 shows the results obtained with the method proposed by Yuan et al. (2015) [25] for the wave excitation forces and the response amplitudes of two ships (Wigley III) in the sway direction, compared to the outcomes from the Wadam (2010) [27] solution.

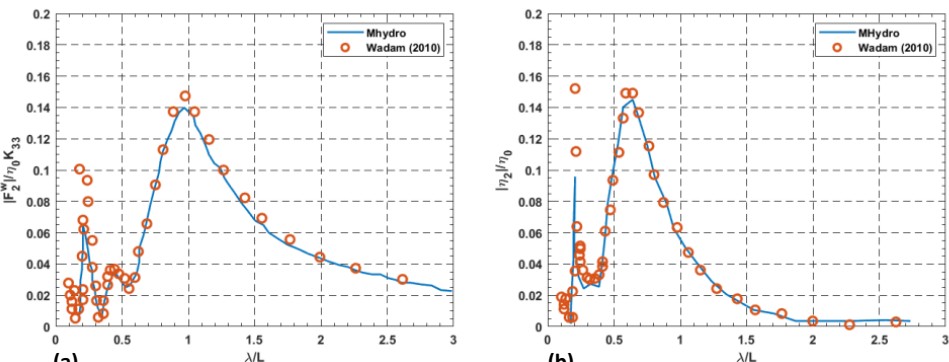

**Figure 19.** Comparison between the method proposed by Yuan et al. [25] and the Wadam solution (Wadam [27]); (**a**) sway force and (**b**) sway motion.

Figure 20 shows the comparison between the sway force on a Wigely hull (Figure 20a) and a rectangular box (Figure 20b) predicted by Yuan et al. [23]. The numerical results of sway force are based on the Green function method (Kashiwagi et al., 2005 [28]) and the experimental values are measured by Kashiwagi et al. (2005) [28].

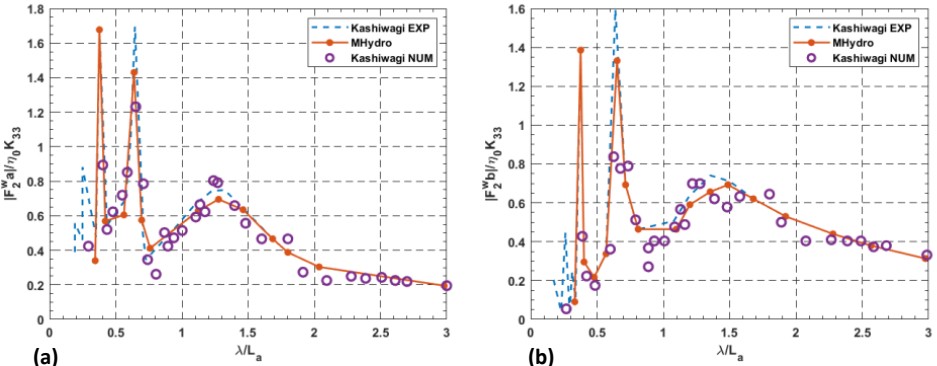

**Figure 20.** Comparison between the method proposed by Yuan et al. [25] and the Wadam solution (Wadam [27]); (**a**) sway force and (**b**) sway motion.

There was a good compliance between the calculated results and those available in the open literature, except for the roll motion; this might be due to the inviscid assumption in the potential flow code. It was found that the hydrodynamic coefficients of the larger ships are determined by her own motion, whereas for the smaller ship, the hydrodynamic coefficients induced by her own oscillation and those due to the presence of a bigger vessel are at the same level.

Yu et al. (2019) [16] suggested that there is a gap in the literature related to the unsteady interactions between vessels travelling at moderate to high speeds. The assumption of the rigid free surface is not reasonable as the wave elevation, due to moving vessels, becomes important. Therefore, in the study, an unsteady potential-flow panel method, called the Unsteady Multi-hull Simple-Source Panel Method (UMSPM), which also integrates the free surface effect, was used. In order to make mesh generation much easier, the free surface mesh was divided into two patches along the centreline between the hulls shown in Figure 21.

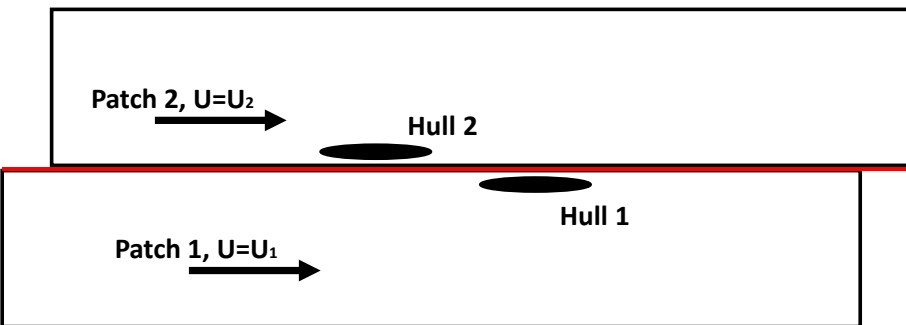

**Figure 21.** Schematic of the free surface. Adapted from Yu et al. [16].

The two patches covering a sufficiently large area of interests move with the corresponding hulls at the same forward speed. Moreover, the free-surface elevation is constant within each panel, so that neither the alignment of the patches nor the re-meshing of the entire domain is required in the time marching. The results demonstrated that during an overtaking manoeuvre, the overtaken vessel blocks the divergent waves generated by the faster one, contrary to when two ships are moving with comparable speeds, and the free surface vertical deformation affects the hydrodynamic loads on both hulls.

## 5. Main Knowledge Gaps

Following the work undertaken in the review of the ship-to-ship interactions in calm waters, the following knowledge gaps have been identified and further work should be undertaken to:

- Close the gap in the literature related to the unsteady interactions between vessels travelling at moderate to high speeds;
- Study the effect of the of the induced waves that large ships impose on tug/traffic boat yawing and heaving, which makes pilot motion control very dangerous;
- Improve measurement techniques to be able to quantify with a high degree of reliability the actual hydrodynamic forces and isolating influences related to the structural-dynamic influences induced by the test setup;
- Quantify the uncertainties associated with the measurement of the loads to allow better validation of numerical tools based on viscous and potential flow approaches;
- Study Vantorre's [2,17] experiments in shallow water in greater detail to further describe the three regression coefficients of dimensionless surge $C_x$, sway $C_y$ and yaw $C_N$, respectively;
- Understand why Vantorre's [2,17] and De Decker's [4] results clearly show that for all cases, the amplitude of the forces and moments differ, and in the case of $C_X$, the trends are different;
- Investigate Lataire et al.'s [1,11,12] parameters $C_{xmax}$, $C_{Ymax}$, $C_{Nmax}$, $C_Y$, $C_T$ and $C_{\xi}$;
- Extend ship interaction studies to include more than parallel heading encounters;
- Evaluate and understand how errors in the numerical methods will be helpful in ship collision avoidance;
- Study the effect of the draught and displacement changes, together with different kinds of hull form vessels that might be used for lightering operations;
- Investigate the two numerical computations, i.e., unsteady Reynolds Averaged Navier-Stokes, RANS method and FHR and the measured forces and moments of the Zou and Larsson's [13] and Sadat-Hosseini et al.'s [23,24] study of lightering operations in shallow water;
- Perform experiments for hull shapes other than that of merchant ships, such as small ferries and high-speed craft.

## 6. Conclusions

The past two decades have witnessed great progress in the research work of ship-to-ship problems. From the experimental studies, there are high-quality benchmark test data on ship-to-ship problems, and those from the 6th MASHCON [29], which focused on overtaking, encountering and passing operations. These benchmark data are essential for the validation of numerical methods. It has also been observed that over the last two decades, more numerical methods have been proposed and validated against these benchmark data. Therefore, numerical predictions are becoming more and more reliable. However, there are still some issues concerning the prediction of the peak values of the interaction forces. In most of the studies, the free-surface effects are not considered. At low-speed manoeuvres in deep water, the effects of the free-surface are not important. However, as the speed increases, the waves generated by the interacting ships should be an important factor for the accurate estimation of the forces, particularly when the ships are manoeuvring in shallow water. This will impose some challenges on numerical modelling. On the other hand, the experimental data on high-speed interactions are not available.

The prediction of the hydrodynamic interaction forces is the first step to address the ship-to-ship problem. To fully simulate the ship-to-ship operations, the manoeuvring motion will also need to be considered, as well as propulsion and rudder control. Because of the unsteady nature of the problem, it creates a great opportunity for future numerical studies.

**Author Contributions:** Conceptualization, C.D.M.-F., T.S., D.V. and M.M.; formal analysis, T.S. and V.V.Z.; investigation, T.S., V.V.Z., D.V. and M.M.; data curation, T.S. and V.V.Z.; writing—original draft preparation, C.D.M.-F., T.S. and V.V.Z.; writing—review and editing, C.D.M.-F., T.S., D.V. and M.M.; visualization, T.S. and V.V.Z.; supervision, C.D.M.-F., D.V. and M.M.; project administration, C.D.M.-F., D.V. and M.M.; funding acquisition, C.D.M.-F., T.S. All authors have read and agreed to the published version of the manuscript.

**Funding:** The presented work has been supported through the collaborative programme of the VENTuRE project. "This project has received funding from the European Union's Horizon 2020 research and innovation programme. Project No. 856887".

**Institutional Review Board Statement:** Not applicable.

**Informed Consent Statement:** Not applicable.

**Data Availability Statement:** Not applicable.

**Conflicts of Interest:** The authors declare no conflict of interest.

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
