# Peer review of "A Review of Ship-to-Ship Interactions in Calm Waters"

_jmse, doi:10.3390/jmse10121856_

Round 1
Reviewer 1 Report
Ship lateral distance has certain influence on navigation efficiency and navigation safety. Due to the large beam of very large ships, the maritime authorities have implemented control measures such as one-way entry and prohibition of overtaking during traffic separation. Secondly, in the busy traffic lanes, the traffic flow is dense, the distance between ships is small. Determining the limit of the distance between ships is helpful to improve the efficiency of navigation. So the research on ship lateral distance is faced with great challenges to a lot of harbour and port.
This manscript has concluded a lot of literatures about experiments, empirical models, and numerical method. However, there are some points canbe done, to improve this manscript.
(1) line 91 & 108 , generally, the longitude position is non-dimensional
(2) line 115, some studies should be cited. So some literatures should be added to prove it.
(3) line 227, in the Empirical Models, Varyani has presnted some results , select one from following as the reference.
Varyani K.S., McGregor R.C., Krishnankutty P., Thavalingam A. New empirical and generic models to predict interaction forces for several ships in encounter and overtaking manoeuvres in a channel. International Shipbuilding Progress, vol. 49, no. 4, pp. 237-262, 2002
Varyani K S, Thavalingam A, Krishnankutty P. New generic mathematical model to predict hydrodynamic interaction effects for overtaking maneuvers in simulators[J]. Journal of Marine Science and Technology, 2004, 9(1):24-31.
Varyani K S, Krishnankutty P. Modification of ship hydrodynamic interaction forces and moment by underwater ship geometry[J]. Ocean Engineering, 2006, 33(8-9):1090-1104.
(4) line 297, the author: "To the best of the authors knowledge, no other methods or formulation are present 297 in open literature.". collect the existed references to prove the detailed presented method. The Varyani's paper has not cited.
(5) line 383 Zhiming Yuan?
(6) line 387 add some picture to show the comparision in Zhiming Yuan paper.
(7) line 412 "5. Main Knowledge Gaps ", Large ship induced wave makes the tug/traffic boat yawing and heaving, and this motion makes pilot much more dangerous. This important in practice.
(8) line 412 "5. Main Knowledge Gaps ", For ship interaction, the parallel heading can not always cover all situation, extending the encountering heading angle will be useful. Or evaluate the existing method, calculate the errors of these methods will be useful. This important in ship collision avoidance.
(9) line 443 Although the 6th MASHCON(2022) has been referred. However there is no literature cited in the manscript.
(10) In the numerical method part, the ship hull mesh and FEM formular can be added.
Author Response
Attached below is the report of Reviewer 1

Reviewer 2 Report
The paper titled "A Review of Ship-to-Ship Interactions in Calm Waters" is recommended for minor revision.
STRENGTHS:
The paper is well prepared.
The paper suits this journal JMSE.
The paper has good flow and organisation.
The paper is well organised and referenced.
The paper has good introduction and discussion.
The authors are well knowledgeable in this field.
The review is detailed but could be improved.
WEAKNESSES:
There should be minor proof reading and some English Language editing.
The authors should also do some discussions on the effect of manoeuvers, stability and coupling in ship-to-ship interactions.
I recommend looking into adding some more references as follows:
https://doi.org/10.1007/s42241-020-0021-5; https://doi.org/10.1016/S1001-6058(13)60426-6; https://doi.org/10.1016/j.oceaneng.2018.11.010; https://doi.org/10.1371/journal.pone.0175840; https://doi.org/10.1016/j.oceaneng.2022.112499;
Also are there recent findings on manoeuvering of ships-to-ship systems of typical FPSOs that could be compared and contrasted to improve the quality of this paper? ALSO, are there studies that show the effect of other loads like ice loads that can affect these systems?
Are there docking issues or cargo transfer studies of these systems that could also be considered in a brief discussion? Are there recommendations of the type of mooring systems and fenders that are applicable for these systems?
Lastly, the introduction should present what calm waters are meant, with some relevant studies to support that.
Author Response
Below is the report to Reviewer 2
